# Factors Associated with Exclusive Breastfeeding during Admission to a Baby-Friendly Hospital Initiative Hospital: A Cross-Sectional Study in Spain

**DOI:** 10.3390/nu16111679

**Published:** 2024-05-29

**Authors:** Cristina Verea-Nuñez, Nuria Novoa-Maciñeiras, Ana Suarez-Casal, Juan Manuel Vazquez-Lago

**Affiliations:** 1Resident Nurse in Pediatrics, Pediatric Service, University Hospital of Santiago de Compostela, Rua da Choupana s/n, 15705 Santiago de Compostela, Spain; cristina.verea.nunez@sergas.es; 2Nurse Specialist in Pediatrics, Hospitalary Unit of Neonatology, University Hospital of Santiago de Compostela, Rua da Choupana s/n, 15705 Santiago de Compostela, Spain; nuria.novoa.macineiras@sergas.es (N.N.-M.); ana.suarez.casal@sergas.es (A.S.-C.); 3Preventive Medicine and Public Health Service, University Hospital of Santiago de Compostela, Rua da Choupana s/n, 15705 Santiago de Compostela, Spain; 4UTAMI, Health Research Institute of Santiago de Compostela (IDIS), 15706 Santiago de Compostela, Spain

**Keywords:** breastfeeding, attitudes, baby-friendly hospital

## Abstract

Background: Breastfeeding is the optimal nourishment for infants and it is recommended that children commence breastfeeding within the first hour of birth and be exclusively breastfed for the initial 6 months of life. Our objective was to determine which factors related to mothers could influence the degree of exclusive breastfeeding during hospitalization, as well as to assess breastfeeding mothers’ attitudes towards breastfeeding. Methods: A multicenter cross-sectional study was undertaken in the healthcare area of Santiago de Compostela, Spain. The necessary variables were collected using a specially designed ad hoc questionnaire. The researcher responsible for recruitment conducted the interviews with the participants. The reduced Iowa Infant Feeding Attitude Scale (IIFAS-s) was employed to gauge maternal attitudes toward feeding their baby. Results: In total, 64 women were studied. The overall score of IIFAS-s (mean ± standard deviation) was 36.95 ± 5.17. A positive attitude towards breastfeeding was therefore observed in our sample. No use of a pacifier by the newborn was associated with a positive attitude for breastfeeding. Having previous children (Ora = 6.40; IC95% 1.26–32.51) and previous experience with breastfeeding (Ora = 6.70; IC95% 1.31–34.27) increased the likelihood of exclusive breastfeeding during admission. Conclusions: In our study, exclusive breastfeeding during hospitalization is associated with having previous children and prior breastfeeding experience.

## 1. Introduction

The World Health Organization (WHO) recognizes breastfeeding (BF) as the optimal nourishment for infants and recommends that children commence breastfeeding within the first hour of birth and be exclusively breastfed for the initial 6 months of life. Subsequently, they should begin consuming safe and suitable complementary foods while continuing breastfeeding for up to two years or beyond [1]. BF not only provides nutritional benefits but also confers psychological and emotional advantages to both the newborn (NB) and the mother [2,3,4,5,6]. Additionally, it contributes to the economic and social well-being of families by promoting better infant health outcomes [6,7,8].

According to the latest National Health Survey in Spain from 2017, breastfeeding was the most prevalent feeding method for babies during the first 6 weeks (73.9%), but it decreased to 63.9% by 3 months. By 6 months, 41.6% of babies were being fed with formula milk, thereby relegating breastfeeding to a secondary position (39%) [9]. This trend is associated with various sociodemographic, clinical, and psychological factors, including maternal insecurity and doubts during the breastfeeding process, as well as the absence of a supportive environment [10,11]. The sociolaboral and cultural shifts of recent decades have negatively impacted breastfeeding rates, with maternal return to work being a primary cause of breastfeeding cessation [10,12,13,14]. Insufficient maternal knowledge about breastfeeding is also a contributing factor to early breastfeeding discontinuation [13,14]. This may partly stem from the lack of or inadequate dissemination of information by health professionals, which in turn can lead to premature breastfeeding cessation [15,16,17]. Furthermore, nursing staff providing care to women in the early postpartum days may also have insufficient knowledge about breastfeeding [18].

Currently, healthcare services are beginning to establish breastfeeding support groups and programs [19,20]. The Initiative for the Humanization of Birth and Breastfeeding Care (BFHI) launched by the WHO and UNICEF aims to encourage hospitals, health services, and particularly maternity wards to adopt practices that protect, promote, and support exclusive breastfeeding from birth [21,22,23]. One of the standards for continuous improvement in these hospitals is that at least 75% of mothers should practice exclusive breastfeeding during hospitalization [24]. Our hospital has been part of the BFHI network since 2015, in Phase 2D since 2020. Hospitals in Phase 2D are required to conduct self-assessments to identify areas for improvement in factors that may influence exclusive breastfeeding [25]. Based on the aforementioned points, it could be expected that a BFHI hospital would have a high proportion of mothers who exclusively breastfeed during hospitalization and continue breastfeeding after discharge. If this is not the case, it is important to understand the factors that may influence these decisions. Our objectives were to determine the percentage of mothers who practiced exclusive BF during hospitalization and which factors related to mothers could influence the degree of exclusive breastfeeding during hospitalization, as well as to assess breastfeeding mothers’ attitudes towards breastfeeding.

## 2. Materials and Methods

### 2.1. Study Design, Population, and Sample

The study was conducted from June 2023 to February 2024 in Galicia, a region in northwest Spain with a population of 2.7 million inhabitants, where breastfeeding abandonment stands at 58.8% within the first year of infant life [26]. To address the study objectives, a multicenter cross-sectional study was undertaken in the healthcare area of Santiago de Compostela, covering a population of 450,000. In 2023, there were 1948 births in this healthcare area.

The sample size was calculated prior to starting the study. Based on the number of births in our healthcare area in 2023, the percentage of exclusive breastfeeding at discharge was estimated at 87.5% [27]. Using a binomial distribution, a sample of at least 62 women was needed, with a 10% absolute error and a 95% confidence level. Once the sample size was determined, primary healthcare centers (PHCs) were randomly selected. From a list of 74 PHCs, 5 were randomly chosen: 2 urban (Concepción Arenal PHC and Vite PHC) and 3 rural (Boqueixón PHC, O Pino PHC, and Touro PHC). Women meeting the selection criteria were invited to participate during routine check-ups through purposive sampling.

### 2.2. Sample Selection and Procedure

To achieve the study objectives, women aged 18 or older, mothers of infants under 12 months who had chosen breastfeeding or started but switched to formula feeding before 6 months, and who gave birth at the clinical hospital of Santiago de Compostela, were randomly selected from the participating PHCs. Participation was offered during contact with the pediatric nurse. Mothers of children older than 12 months or those opting for formula feeding were excluded.

Data were collected using a specific data collection notebook comprising sections on the sociodemographic variables of women, variables related to children, and variables related to breastfeeding, including type of breastfeeding during admission and discharge, support and information on breastfeeding during admission and follow-up in the HC, and family support for breastfeeding.

Additionally, the Reduced Iowa Infant Feeding Attitude Scale (IIFAS-s) was employed to gauge maternal attitudes toward feeding their baby, validated for the Spanish population [28,29].

IIFAS-s scale consists of 9 items, each rated from 1 (completely disagree) to 5 (completely agree) on a Likert scale. It has a unidimensional structure with adequate reliability and validity results [27,28]. This scale also serves as a predictive indicator of the choice of feeding method (breastfeeding, formula, or mixed) and breastfeeding duration. It assesses maternal attitudes towards infant feeding pre- and postpartum, identifying women at risk of not initiating breastfeeding. Item scores were grouped into three categories: disagree/positive towards formula feeding (Scores 1 and 2), neutral (Score 3), and agree/positive towards breastfeeding (Scores 4 and 5). Based on the average score, values below 18 classify the sample as “positive attitude towards formula feeding”; values between 18 and 36 as “neutral attitude”; and values above 36 as “positive attitude towards breastfeeding”.

The IIFAS-s questionnaire was used because it is a valid and reliable tool for measuring mothers’ attitudes towards breastfeeding. This allows us to estimate, based on the scores of this questionnaire, the effect that a positive attitude towards breastfeeding has on maintaining exclusive breastfeeding during hospitalization. The IIFAS-s was administered either on paper or online via a QR code, voluntarily and anonymously, through the nurse or midwife.

### 2.3. Ethical and Legal Considerations

The study was approved by the Territorial Committee of Ethics in Research of Santiago-Lugo (registration code: 2023/199), ensuring informed consent from participants.

### 2.4. Variables and Statistical Analysis

Sociodemographic variables (mother’s age, child’s age, educational level, economic status, type of delivery, social and family support, return to work) and hospitalization-related variables (previous breastfeeding experience, skin-to-skin contact, child’s admission, information received about breastfeeding, type of breastfeeding during admission) were collected.

Qualitative variables were presented as numbers and percentages, and quantitative variables as the mean and standard deviation or median and interquartile range.

Bivariate analyses explored the relationship between maternal characteristics or hospitalization-related variables and those measured through the IIFAS-s. Logistic regression models calculated crude and adjusted odds ratios (ORs), with confounding variables included based on significance in the bivariate analysis (*p* < 0.1). All analyses adhered to a 95% confidence level and significance at *p* < 0.05.

## 3. Results

### 3.1. Sample Description

A total of 64 women were studied, with a participation rate of 100%. All women offered participation in the study accepted. The mean age of the mothers was 36.6 ± 4.1 years, with 45 (70.3%) being ≥35 years old. The mean age of the children was 6.3 ± 3.6 months.

Nine (14.1%) newborns were admitted to the hospital’s neonatology unit at the time of birth. Further characteristics of the participating women can be seen in Table 1.

Regarding the assistance received from professionals during their hospitalization, 41 (64.1%) women considered it good, while 23 (35.9%) considered it improvable. Regarding the information received during hospitalization about breastfeeding, 36 women (56.3%) considered it good, while 28 (43.7%) considered it improvable.

In total, 30 (46.9%) of the women had received information about breastfeeding at their PHC from the pediatric nurse, 20 (31.3%) considered they did not need it, and 14 (21.9%) did not receive any information at their PHC. Regarding the level of satisfaction received, 5 (7.8%) women declared not being satisfied at all, 4 (6.3%) declared being somewhat satisfied, 6 (9.4%) fairly satisfied, and 15 (23.4%) very satisfied.

Despite receiving information about breastfeeding in the hospital and PHC, 31 (48.4%) had contacted breastfeeding support groups/counseling, and 58 (90.6%) declared having good family support for breastfeeding.

A high percentage of women did not receive assistance from healthcare professionals regarding breastfeeding during hospitalization. It was also observed that the women who did receive assistance and information about breastfeeding during hospitalization or in their PHC found that the information could be improved. These results may explain why around half of the sample (48.4%) had contacted breastfeeding support groups/counseling.

### 3.2. Attitudes towards Breastfeeding

Table 2 shows women’s attitudes towards breastfeeding through the IIFAS-s. The overall score (mean ± standard deviation) of the test was 36.95 ± 5.17. Items 5 and 6 had the lowest and highest scores, respectively. In our sample, we did not observe any women who fell into the negative attitude towards breastfeeding category. Twenty-one (32.8%) women exhibited a neutral attitude, and forty-three (67.2%) showed a positive attitude towards breastfeeding.

It was observed that the mean score for each item is above three, indicating that the studied sample has a positive attitude towards breastfeeding, with Items 3, 4, 5, and 8 being closest to neutrality. Additionally, 72.57% of the women scored the IIFAS-s items with values of four or five, demonstrating a positive attitude towards breastfeeding, compared to 8.85% of the women who scored the items with values of one or two, indicating a positive attitude towards formula feeding (Table 2).

Women whose newborns do not use pacifiers show higher scores on the IIFAS-s, indicating a more favorable attitude towards breastfeeding (see Table 3). Conversely, no differences are observed between women’s demographic variables and the mean scores of the IIFAS-s.

### 3.3. Factors Influencing Exclusive Breastfeeding during Hospitalization

The following table (Table 4) shows factors associated with the woman or certain hospitalization characteristics that may influence the implementation of exclusive breastfeeding during hospitalization. It can be observed that having previous children, prior experience with breastfeeding, and the newborn not being admitted to the neonatology unit increase the likelihood of exclusive breastfeeding during admission.

When adjusting the odds ratio (OR) associated with these variables for possible confounding variables, having previous children and prior experience with breastfeeding remain associated with exclusive breastfeeding during hospitalization. Among the most important findings, it is worth highlighting that having previous children increases the probability of exclusive breastfeeding during hospitalization by 6.4 times. Additionally, prior breastfeeding experience increases this probability by 6.7 times.

## 4. Discussion

To our knowledge, this is the first study to evaluate factors associated with exclusive breastfeeding, especially during hospital admission. Our results show that factors such as having previous children or prior experience with breastfeeding increase the likelihood of exclusive breastfeeding during subsequent births. Studies have linked multiparity with a positive association with breastfeeding duration [29]. Additionally, other studies have confirmed that previous breastfeeding experiences, unsuccessful attempts at breastfeeding, and the inability to breastfeed the first child have been associated with lower breastfeeding initiation rates in subsequent children [30,31,32].

The results of our study show a percentage of women exclusively breastfeeding during admission of 73.4%, with 75% being the sentinel indicator for the rate of exclusive breastfeeding at discharge for BFHI accreditation [33]. This study identified that the number of women breastfeeding exclusively post-discharge increased by seven respondents (10% more), possibly explained by the role of the primary care pediatric nurse or the mother’s contact with breastfeeding support groups. In our sample, we observed that a high percentage of mothers contacted breastfeeding support groups. Despite giving birth in a BFHI hospital, 35.9% of the women did not receive assistance or information about breastfeeding from the healthcare professionals who cared for them during their stay. Among those who did receive assistance, a high percentage considered it could be improved. A similar situation occurred at the primary healthcare level. This could be explained by the need to improve training activities for healthcare professionals. Evidence demonstrates that interventions to support breastfeeding in primary care have a positive effect on breastfeeding rates, duration, or exclusive maintenance [34]. A systematic review by Balogun et al. asserts that the rate of breastfeeding initiation improves among women who receive breastfeeding education and support led by healthcare professionals compared to those who receive standard care [35].

Skin-to-skin contact, performed by 81.3% of participants, appears to be beneficial for breastfeeding in the short and long term, as shown in a systematic review that observed improvements in both breastfeeding status and duration [32]. Regarding factors influencing exclusive breastfeeding during hospitalization, we observed that if the child is not admitted to the neonatal unit, there is an increased probability of establishing exclusive breastfeeding during admission, as well as having previous experience with exclusive breastfeeding. This is consistent with studies demonstrating that rooming-in mother/child in neonatal units increases the probability of successful exclusive breastfeeding [36,37,38]. Two areas have been identified where specific interventions could be focused on the healthcare staff who care for women during their hospital stay. By concentrating training interventions on promoting skin-to-skin contact within the first 30 min, as well as equipping hospitals with more humanized and open neonatal units that encourage activities like the kangaroo mother care technique and skin-to-skin contact, the likelihood that mothers will choose exclusive breastfeeding during their stay would increase. Additionally, the duration of breastfeeding is likely to extend once they are discharged from the hospital. There is evidence that addressing the two identified areas would achieve these goals [39,40].

The total score of the IIFAS-s scale in our study does not differ from the available evidence, where it can be observed that women present positive attitudes towards exclusive breastfeeding, especially during pregnancy and hospital admission [41,42]. In this study, we did not observe any women who fell into the negative attitude towards breastfeeding category and 67.2% showed a positive attitude towards breastfeeding. This result is much higher than those found in previous studies [43,44]. It should be taken into account when interpreting these results that the women who participated did so after their hospital stay, while they were being followed up by pediatric nurses at their PHCs. This may be influenced by the fact that the women received advice at that time from their pediatric nurse and breastfeeding support groups. Regarding the results extracted from the IIFAS-s scale, it can be observed that mothers’ attitudes towards breastfeeding through the IIFAS-s scale do not show statistically significant differences by demographic factor. Concerning pacifier use, this was systematically questioned since numerous studies demonstrate that pacifier use is related to a lower rate of exclusive breastfeeding, although some demonstrate the opposite [20]. Our data reflect that women whose newborns do not use pacifiers show higher scores on the IIFAS-s, indicating a more favorable attitude towards breastfeeding. Additionally, it is noteworthy that only 50% of the surveyed mothers reported that breastfed babies are healthier than formula-fed babies, when no literature has been found to demonstrate otherwise.

As a strength of this study, we would like to highlight the survey as a cost-effective and efficient tool for obtaining data: its accessibility, ease of use, and availability in both paper and QR code formats have allowed us to reach the target population in a short period. Additionally, the IIFAS-s scale is considered a good predictor of attitudes towards initiating exclusive breastfeeding, although not as a predictor of maintaining exclusive breastfeeding during hospital admission [28]. By using these two methods in this study, we consider that we used the appropriate tool to obtain a representative picture of the attitudes and characteristics of our group.

Another strength of this study is that it allows us to identify areas for improvement or gaps to target and focus interventions on, both for mothers and healthcare professionals, as well as the healthcare system. This is consistent with similar studies [45].

Regarding the study’s limitations, it is worth mentioning the inherent limitations of a cross-sectional design, although our results serve to generate hypotheses on the topic of the work. On the other hand, the achieved sample size may not be sufficient to provide high power to our results. It would be necessary to carry out studies with prospective designs to corroborate our results.

## 5. Conclusions

In our study, exclusive breastfeeding during hospitalization is associated with having previous children and prior breastfeeding experience. It also appears to be linked to the likelihood of the newborn not being admitted to neonatal units. This association is influenced by the mother’s age, breastfeeding information provided during hospitalization, and skin-to-skin contact within the first 30 min. Exclusive breastfeeding during hospitalization could be improved by increasing healthcare staff training and encouraging their involvement in achieving goals according to the BFHI criteria. Future research with prospective designs is needed to measure the effect of multifaceted interventions focused on our findings, in order to estimate the association with exclusive breastfeeding during hospitalization and its duration after discharge.

## Figures and Tables

**Table 1 nutrients-16-01679-t001:** Characteristics of women (*n* = 64).

Variable	n	%
**Age**		
<35 years	19	20.7
≥35 years	45	70.3
**Type of population**		
Urban	45	70.3
Rural	19	29.7
**Family income**		
<18.000 €/año	12	18.8
≥18.000 €/año	52	81.3
**Type of childbirth**		
Cesarean	15	23.4
Vaginal	49	76.6
**Previous children**		
Yes	27	42.2
No	37	57.8
**Use of pacifier by newborn**		
Yes	26	40.6
No	38	59.4
**Previous experience in breastfeeding**		
Yes	27	42.2
No	37	57.8
**Skin-to-skin contact in first 30 min**		
Yes	52	81.3
No	12	18.8
**Exclusive breastfeeding during hospital admission**		
Yes	47	73.4
No	17	26.6
**Exclusive breastfeeding at hospital discharge**		
Yes	54	84.4
No	10	15.6

**Table 2 nutrients-16-01679-t002:** Women’s attitudes towards breastfeeding (IIFAS-s).

Ítem. Variable (a)	M	SD	Agreement (%)	Neutral (%)	Disagreement (%)
1. Formula feeding is more convenient than breastfeeding (b)	4.63	0.84	92.2	4.7	3.1
2. Breastfeeding strengthens the bond between mother and child	4.63	0.84	92.2	4.7	3.1
3. Formula feeding is the best option if the mother intends to work outside the home (b)	3.8	1.04	64.1	23.4	12.5
4. Mothers who do not breastfeed miss out on one of the best experiences of motherhood	3.53	1.19	50	34.4	15.6
5. Breastfed babies are healthier than formula-fed babies	3.44	1.27	50	29.7	20.3
6. Breast milk is the ideal food for the baby	4.78	0.58	95.3	3.1	1.6
7. Breast milk is more easily digested than formula milk	4.31	0.94	75	23.4	1.6
8. Formula milk is as healthy for the baby as breast milk (b)	3.7	1.11	57.8	28.1	14.1
9. Breastfeeding your baby is more convenient than not doing so	4.14	1.14	76.6	15.6	7.8
Total	36.95	5.17	72.6	18.6	8.8

(a) Participants (*n* = 64) were asked if they agreed with each statement on a 5-point Likert scale ranging from 1 (strongly disagree) to 5 (strongly agree). These scores were then grouped into the following three categories: disagree/positive towards formula feeding (Scores 1 and 2), neutral (Score 3), and agree/positive towards breastfeeding (Scores 4 and 5). (b) These items were reversed when calculating the score. M: mean. SD: standard deviation.

**Table 3 nutrients-16-01679-t003:** Differences in attitudes towards breastfeeding by demographic factor, as determined by scores on the IIFAS-s scale. Higher IIFAS-s scores reflect more positive attitudes towards breastfeeding.

Factor (a)	Category	Mean Score (SD)	*p*
Mother’s age	<35 years	35.32 (6.05)	0.145
	≥35 years	37.64 (4.66)	
Type of population	Rural	36.58 (6.24)	0.741
	Urban	37.11 (4.72)	
Family incomes	<18,000 €/year	37.08 (5.73)	0.924
	≥18,000 €/year	36.92 (5.10)	
Type of childbirth	Cesarean	37.13 (4.44)	0.879
	Vaginal	36.90 (5.42)	
Previous children	Yes	37.30 (4.58)	0.537
	No	36.48 (5.94)	
Use of pacifier by the newborn	Yes	35.35 (5.61)	0.039 *
	No	38.05 (4.61)	
Previous breastfeeding experience	Yes	36.67 (6.01)	0.708
	No	37.16 (4.54)	
Skin-to-skin contact during the first 30 min	Yes	36.92 (5.34)	0.924
	No	37.08 (4.56)	
Admission of the newborn to neonatology	Yes	37.22 (4.94)	0.868
	No	36.91 (5.25)	
Exclusive breastfeeding at hospital discharge	Yes	37.31 (5.14)	0.196
	No	35.00 (5.14)	
Perception of proper assistance from healthcare professional during admission	Yes	36.34 (5.26)	0.209
	No	38.04 (4.94)	
Perception of proper information about breastfeeding from healthcare professional during admission	Yes	35.86 (5.21)	0.055
	No	38.36 (4.86)	
Contact with breastfeeding support groups	Yes	37.84 (5.34)	0.186
	No	36.12 (4.94)	
Family support for breastfeeding	Yes	36.91 (4.99)	0.852
	No	37.33 (7.29)	

(a) n = 64. SD: Standard deviation. * *p* < 0.05 test *t*-Student.

**Table 4 nutrients-16-01679-t004:** Factors influencing exclusive breastfeeding during hospitalization. n = 64.

Exclusive Breastfeeding during Hospital Admission				
Factor	YES n (%)	NO n (%)	ORc (IC 95%)	ORa (IC 95%)
Mother’s age ≥ 35 years	36 (76.6)	9 (52.9)	2.91 (0.91–9.35)	1.89 (0.50–7.06)
Urban population	31 (66.0)	14 (82.4)	2.41 (0.60–9.62)	3.21 (0.55–18.82)
Incomes ≥ 18.000 €/year	39 (83.0)	13 (76.5)	1.50 (0.39–5.81)	2.08 (0.43–10.10)
Vaginal childbirth	36 (76.6)	13 (76.5)	1.01 (0.27–3.73)	1.34 (0.28–6.41)
≥1 previous children	25 (53v2)	2 (11.8)	8.52 (1.75–41.49)	6.40 (1.26–32.51)
Previous breastfeeding experience	25 (53.2)	2 (11.8)	8.52 (1.75–41.49)	6.70 (1.31–34.27)
Skin-to-skin contact during the first 30 min	40 (85.1)	12 (70.6)	2.38 (0.64–8.88)	1.18 (0.22–6.19)
No admission of the newborn to neonatology	43 (91.5)	12 (70.6)	4.48 (1.04–19.33)	3.41 (0.68–17.02)
Use of pacifier by the newborn	20 (42.6)	6 (35.3)	1.36 (0.43–4.29)	1.71 (0.45–6.50)
Perception of proper assistance from healthcare professional during admission	31 (66.0)	10 (58.8)	1.36 (0.43–4.24)	1.67 (0.45–6.20)
Perception of proper information about breastfeeding from healthcare professional during admission	29 (61.7)	7 (41.2)	2.30 (0.74–7.13)	1.92 (0.54–6.86)
Contact with breastfeeding support groups	24 (51.1)	7 (41.2)	1.49 (0.48–4.58)	1.59 (0.46–5.57)
Family support for breastfeeding	43 (91.5)	15 (88.2)	1.43 (0.24–8.64)	1.04 (0.14–7.87)

ORc: Crude odds ratio. ORa: Adjusted odds ratio. Adjusted for the following variables: mother’s age, information regarding breastfeeding during hospitalization, performing skin-to-skin contact in the first 30 min.

## Data Availability

Data availability is under petition.

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
