# Peer review of "Factors Associated with Exclusive Breastfeeding during Admission to a Baby-Friendly Hospital Initiative Hospital: A Cross-Sectional Study in Spain"

_nutrients, 2024, doi:10.3390/nu16111679_

Round 1

Reviewer 1 Report

Comments and Suggestions for Authors

The manuscript entitled ‘Factors associated with exclusive breastfeeding during admission to a BFHI hospital. A cross-sectional study in Spain' presents important findings a regarding factors influencing women's choice of exclusive breastfeeding during perinatal hospitalisation. Exclusive breastfeeding of newborns and young children (up to 6 months of age) is an issue of scientifically proven and widely described importance. In contrast, it is known that a significant percentage of women do not take this action. This, of course, has various reasons and it is this area that the authors have addressed in the study described in the manuscript under review.

Below are the comments that I consider necessary to include in the preparation of a revised version of the manuscript. I have included my comments and observations according to the order of each section, not according to their importance.

1. Please remove the abbreviation ‘BFHI’ from the title of the manuscript and replace it with the full name.

2. Abstract section:

a) Please complete the name of the method (and/or tool) to investigate the factors influencing exclusive breastfeeding during hospitalisation, which the authors wrote about in lines19-20.

b) The sentence in lines 24-25 a regarding the mean and SD of the IIFAS-s scale score (‘The overall score of IIFAS-s (mean ± standard deviation) was 36.95 ± 5.17’) should be completed with an interpretation of the posted numerical score. The numerical score itself is only useful for those familiar with how to interpret these results.

c) This section should contain a short conclusion directly derived from the results obtained in the study conducted. At present, in the Conclusions sub-section, the authors have included a very vague sentence that is more of a recommendation.

3. Introduction section:

a) Comment on the sentence at lines 54-55: ‘This may partly stem from the lack of or inadequate dissemination of information by nursing staff, which in turn can lead to premature breastfeeding cessation [17-19].’ The authors appear to be placing the onus of responsibility for premature cessation of neonatal breastfeeding by mothers solely on nursing staff. This seems an unfair and unwarranted assumption. It would be beneficial to ascertain whether other medical staff lack the capacity to act in this area.

b) Please explain the abbreviation ‘IHAN’ for the first time in line 66.

4. Materials and Methods section:

a) Please provide an explanation of the abbreviation ‘HC’ for the first time on line 87.

b) For the description concerning the IIFAS scale, please indicate how the results of this study were interpreted, i.e. which cut-off point was used. Please also provide a detailed account of the specific results and their indicative value.

5. Results section:

a) The authors state that they surveyed 64 women. However, it is unclear whether this was the number of subjects they intended to survey. To clarify this point, please provide further information regarding the selection of the size of the study sample and the process of recruiting female study participants.

b) In Table 1, it is unnecessary to repeat '(n=64)' in each row with the parameters characterising the study group. It is sufficient to state this value in the heading of this table.

c) In Table 1, please provide an explanation of the abbreviation ‘NB’.

d) In Table 1, the expression ‘kin-to-skin contact in first 30 minutes (n=64)’ requires correction. The correct expression is ‘skin-to-skin contact in first 30 minutes’.

6. The entire discussion section must be completed with references to the relevant literature. The authors in the text of this section refer to the results of studies by other authors, yet fail to fulfil their obligation to insert a reference to the literature items describing these studies.

7. Conclusions section: While the authors have included some valuable statements, this section should primarily present a concise conclusion derived directly from the study and its results.

8. References section: This section contains a considerable number of literature items in a language other than English (Spanish). This fact represents a significant limitation, as it makes it challenging, if not impossible, to assess the relevance of the literature items used.

Author Response

REVIEWER 1.

The manuscript entitled ‘Factors associated with exclusive breastfeeding during admission to a BFHI hospital. A cross-sectional study in Spain' presents important findings a regarding factors influencing women's choice of exclusive breastfeeding during perinatal hospitalisation. Exclusive breastfeeding of newborns and young children (up to 6 months of age) is an issue of scientifically proven and widely described importance. In contrast, it is known that a significant percentage of women do not take this action. This, of course, has various reasons and it is this area that the authors have addressed in the study described in the manuscript under review.

Dear Reviewer,

Thank you very much for your kind words. They encourage us to continue working in this direction. We consider your feedback very important, as it allows us to know that our theoretical framework has been correctly constructed and that our objective, focused on the reasons that may influence mothers' decisions not to breastfeed their newborns, has been understood. Thank you very much.

Below are the comments that I consider necessary to include in the preparation of a revised version of the manuscript. I have included my comments and observations according to the order of each section, not according to their importance.

Thank you. Your comments have been evaluated by the research team and addressed in the manuscript. We hope that our responses help clarify your concerns.

1. Please remove the abbreviation ‘BFHI’ from the title of the manuscript and replace it with the full name.

Ok. The abbreviation “IHAN” has been removed and replaced with the full name. In this new version of the manuscript, the title is “Factors Associated with Exclusive Breastfeeding During Admission to a Baby-Friendly Hospital Initiative Hospital: A Cross-Sectional Study in Spain.”

2. Abstract section:

a) Please complete the name of the method (and/or tool) to investigate the factors influencing exclusive breastfeeding during hospitalisation, which the authors wrote about in lines19-20.

Sociodemographic variables and those related to potential factors that could influence exclusive breastfeeding during hospitalization were collected using a specially designed ad hoc data collection notebook. The researcher responsible for recruiting mothers in the primary health centers conducted interviews with the participating women to collect the data defined in the notebook. This information is added to the methods section in the abstract. In the new version of the manuscript, this information is presented as follows: “The necessary variables were collected using a specially designed ad hoc questionnaire. The researcher responsible for recruitment conducted the interviews with the participants.”

b) The sentence in lines 24-25 a regarding the mean and SD of the IIFAS-s scale score (‘The overall score of IIFAS-s (mean ± standard deviation) was 36.95 ± 5.17’) should be completed with an interpretation of the posted numerical score. The numerical score itself is only useful for those familiar with how to interpret these results.

Thank you very much. We agree with your assessment. The IIFAS-s scale consists of 9 items, each rated from 1 (completely disagree) to 5 (completely agree) on a Likert scale. It has a unidimensional structure, and the reliability and validity results are adequate. This scale also serves as a predictive indicator of the chosen feeding method (breastfeeding, formula, or mixed) and the duration of breastfeeding. It evaluates maternal attitudes towards infant feeding pre- and post-partum and identifies women at risk of not initiating breastfeeding. The scores for each item were grouped into the following three categories: disagree/positive towards formula feeding (scores 1 and 2), neutral (score 3), and agree/positive towards breastfeeding (scores 4 and 5).

Based on the mean, values less than 18 classify the sample as "positive attitude towards formula feeding"; values between 18 and 36 classify it as "neutral attitude"; and values greater than 36 as "positive attitude towards breastfeeding." With a mean of 36.95 ± 5.17, this indicates that our sample is classified as having a "positive attitude towards breastfeeding." A brief explanatory sentence has been added to the new version of the manuscript: “A positive attitude towards breastfeeding is therefore observed in our sample.”

c) This section should contain a short conclusion directly derived from the results obtained in the study conducted. At present, in the Conclusions sub-section, the authors have included a very vague sentence that is more of a recommendation.

Thank you very much. We agree with the assessment. It has been corrected in the new version of the manuscript. It now reads: “In our study, exclusive breastfeeding during hospitalization is associated with having previous children and prior breastfeeding experience.”

3. Introduction section:

a) Comment on the sentence at lines 54-55: ‘This may partly stem from the lack of or inadequate dissemination of information by nursing staff, which in turn can lead to premature breastfeeding cessation [17-19].’ The authors appear to be placing the onus of responsibility for premature cessation of neonatal breastfeeding by mothers solely on nursing staff. This seems an unfair and unwarranted assumption. It would be beneficial to ascertain whether other medical staff lack the capacity to act in this area.

It is not the intention of the research team to place sole responsibility on nursing staff for the premature cessation of breastfeeding. We referred to these professionals because the study was conducted on them, and focusing on them allowed us to better define our theoretical framework. In Spain, the responsibility of promoting breastfeeding falls almost exclusively on nursing staff, both in hospital and community settings; to a lesser extent, this responsibility also falls on pediatricians.

However, we agree with the reviewer that the phrasing, as it stands, appears to place the issue of premature breastfeeding cessation solely on nursing. This is not the case. In fact, the systematic reviews on which we base this assumption evaluate healthcare professionals as a whole, not just nurses. We have modified the phrase the reviewer referred to.

In the current version of the text, it reads as follows: “This may partly stem from the lack of or inadequate dissemination of information by health professionals, which in turn can lead to premature breastfeeding cessation [17-19].”

b) Please explain the abbreviation ‘IHAN’ for the first time in line 66.

IHAN is the Spanish abbreviation that means the same as BFHI. We apologize to the reviewers for not using the appropriate terminology in English. This has now been corrected throughout the entire manuscript. The terminology has been updated on line 66 and on line 218 (second paragraph of the discussion). Thank you very much for identifying this terminology error.

4. Materials and Methods section:

a) Please provide an explanation of the abbreviation ‘HC’ for the first time on line 87.

This is an error. The abbreviation HC that appears on line 87 should actually be PHCs, which is explained in the previous paragraph, referring to Primary Health Centers (PHCs). Thank you again for identifying and communicating this error.

b) For the description concerning the IIFAS scale, please indicate how the results of this study were interpreted, i.e. which cut-off point was used.

Thank you very much for your input. Explaining the characteristics of the reduced IIFAS-s scale we used in more detail will increase the validity and reproducibility of our study. As we explained earlier, the IIFAS-s scale consists of 9 items, each rated from 1 (completely disagree) to 5 (completely agree) on a Likert scale. It has a unidimensional structure, and the reliability and validity results are adequate. This scale also serves as a predictive indicator of the chosen feeding method (breastfeeding, formula, or mixed) and the duration of breastfeeding. It evaluates maternal attitudes towards infant feeding pre- and post-partum and identifies women at risk of not initiating breastfeeding. The scores for each item were grouped into the following three categories: disagree/positive towards formula feeding (scores 1 and 2), neutral (score 3), and agree/positive towards breastfeeding (scores 4 and 5).

Based on the mean, values less than 18 classify the sample as having a “positive attitude towards formula feeding”; values between 18 and 36 classify it as having a “neutral attitude”; and values greater than 36 classify it as having a “positive attitude towards breastfeeding.”

This information is incorporated into the materials and methods section. In the new version of the manuscript, it reads as follows: “The IIFAS-s scale consists of 9 items, each rated from 1 (completely disagree) to 5 (completely agree) on a Likert scale. It has a unidimensional structure with adequate reliability and validity results [29,30]. This scale also serves as a predictive indicator of the choice of feeding method (breastfeeding, formula, or mixed) and breastfeeding duration. It assesses maternal attitudes towards infant feeding pre- and post-partum, identifying women at risk of not initiating breastfeeding. Item scores were grouped into three categories: disagree/positive towards formula feeding (scores 1 and 2), neutral (score 3), and agree/positive towards breastfeeding (scores 4 and 5). Based on the average score, values below 18 classify the sample as "positive attitude towards formula feeding"; values between 18 and 36 as "neutral attitude"; and values above 36 as "positive attitude towards breastfeeding.”

Please also provide a detailed account of the specific results and their indicative value.

Thank you very much. We accept the suggestion. This enhances the value of the results section. In the new version of the text, it appears as follows, in the results section: “It was observed that the mean score for each item is above 3, indicating that the studied sample has a positive attitude towards breastfeeding, with items 3, 4, 5, and 8 being closest to neutrality. Additionally, 72.57% of the women scored the IIFAS-s items with values of 4 or 5, demonstrating a positive attitude towards breastfeeding, compared to 8.85% of the women who scored the items with values of 1 or 2, indicating a positive attitude towards formula feeding (Table 2).”

5. Results section:

a) The authors state that they surveyed 64 women. However, it is unclear whether this was the number of subjects they intended to survey. To clarify this point, please provide further information regarding the selection of the size of the study sample and the process of recruiting female study participants.

Thank you very much. The sample size was calculated a priori before initiating the study. For sample size calculation, considering that there were 1948 births in our healthcare area during the year 2023, and assuming the percentage of exclusive breastfeeding at discharge was estimated (based on previous literature) to be 87.5% [1], and using calculation based on a binomial distribution, a sample size of at least 62 women was required to estimate a proportion of 87.5%, with an absolute error of 10%, and for a confidence level of 95%. In our study, we recruited a total of 64 women.

Once we set the sample size a priori, we proceeded to select the primary healthcare centers (PHCs) that would participate. Starting from a list of all primary healthcare centers in the healthcare area (74 in total), we randomly selected 5 primary healthcare centers to recruit participants. The five selected centers mentioned in the manuscript were Concepción Arenal PHC and Vite PHC (urban), and Boqueixón PHC, O Pino PHC, and Touro PHC (rural).

After selecting the participating healthcare centers, women attending a routine check-up at the center and meeting the selection criteria were invited to participate in the study. All 64 women invited agreed to participate; there were no refusals. Recruitment was stopped upon reaching 64 participants, as we had reached the target sample size.

This information on sample size is included in the materials and methods section in the new version of the text. In the revised manuscript, it is presented as follows: “The sample size was calculated prior to starting the study. Based on the number of births in our healthcare area in 2023, the percentage of exclusive breastfeeding at discharge was estimated at 87.5% [25]. Using a binomial distribution, a sample of at least 62 women was needed, with a 10% absolute error and a 95% confidence level. Once the sample size was determined, primary healthcare centers (PHCs) were randomly selected. From a list of 74 PHCs, 5 were randomly chosen: 2 urban (Concepción Arenal PHC and Vite PHC) and 3 rural (Boqueixón PHC, O Pino PHC, and Touro PHC). Women meeting the selection criteria were invited to participate during routine check-ups through purposive sampling.”

[1] (ref. 25 in manuscript) García García N, Fernández Gutiérrez P. Conocimientos y actitudes de las madres ante la lactancia materna en un hospital IHAN. Metas Enferm 2018; 21(1):50-8. doi:10.35667/MetasEnf.2019.21.1003081174

b) In Table 1, it is unnecessary to repeat '(n=64)' in each row with the parameters characterising the study group. It is sufficient to state this value in the heading of this table.

Thank you very much. This has been corrected in the new version of the manuscript.

c) In Table 1, please provide an explanation of the abbreviation ‘NB’.

Thank you for the observation. The abbreviation "NB" is explained in the first paragraph of the introduction (lines 41-42): "BF not only provides nutritional benefits but also confers psychological and emotional advantages to both the newborn (NB) and the mother [2-6]."

However, we will proceed to replace the abbreviation "NB" with the full term in Table 1. We understand that the reader may not recall the meaning of this abbreviation when seeing it again in Table 1. This change will improve readability. Thank you for your input.

d) In Table 1, the expression ‘kin-to-skin contact in first 30 minutes (n=64)’ requires correction. The correct expression is ‘skin-to-skin contact in first 30 minutes’.

Thank you for the observation. We will proceed to correct it. The revised version of the manuscript reflects this correction. Thank you.

6. The entire discussion section must be completed with references to the relevant literature. The authors in the text of this section refer to the results of studies by other authors, yet fail to fulfil their obligation to insert a reference to the literature items describing these studies.

Thank you for the observation. There was an error in handling the Refworks software. It has now been corrected. All references from the discussion section are included in the new version of the manuscript. Thank you.

7. Conclusions section: While the authors have included some valuable statements, this section should primarily present a concise conclusion derived directly from the study and its results.

Thank you very much for your observation. Following the recommendation from your review, the conclusions section has been modified. In the new version of the manuscript, it reads as follows: “In our study, exclusive breastfeeding during hospitalization is associated with having previous children and prior breastfeeding experience. It also appears to be linked to the likelihood of the newborn not being admitted to neonatal units. This association is influenced by the mother's age, breastfeeding information provided during hospitalization, and skin-to-skin contact within the first 30 minutes. Exclusive breastfeeding during hospitalization could be improved by increasing healthcare staff training and encouraging their involvement in achieving goals according to BFHI criteria. Future research with prospective designs is needed to measure the effect of multifaceted interventions focused on our findings, in order to estimate the association with exclusive breastfeeding during hospitalization and its duration after discharge.”

8. References section: This section contains a considerable number of literature items in a language other than English (Spanish). This fact represents a significant limitation, as it makes it challenging, if not impossible, to assess the relevance of the literature items used.

Thank you very much for your comment. References have been corrected as you suggested. Thank you.

Due to all the changes prompted by the reviewers' suggestions, new bibliographic references had to be included. It are:

42. Song JT, Kinshella MW, Kawaza K, Goldfarb DM. Neonatal Intensive Care Unit Interventions to Improve Breastfeeding Rates at Discharge Among Preterm and Low Birth Weight Infants: A Systematic Review and Meta-Analysis. Breastfeed Med. 2023;18(2):97-106. doi:10.1089/bfm.2022.0151.

43. Renfrew MJ, Dyson L, McCormick F, Misso K, Stenhouse E, King SE, Williams AF. Breastfeeding promotion for infants in neonatal units: a systematic review. Child Care Health Dev. 2010;36(2):165-78. doi:10.1111/j.1365-2214.2009.01018.x.

46. Alkhaldi SM, Al-Kuran O, AlAdwan MM, Dabbah TA, Dalky HF, Badran E. Determinants of breastfeeding attitudes of mothers in Jordan: A cross-sectional study. PLoS One. 2023 May 5;18(5):e0285436. doi:10.1371/journal.pone.0285436.

47. Han FL, Ho YJ, McGrath JM. The influence of breastfeeding attitudes on breastfeeding behavior of postpartum women and their spouses. Heliyon. 2023;9(3):e13987. doi:10.1016/j.heliyon.2023.e13987.

48. Al-Thubaity DD, Alshahrani MA, Elgzar WT, Ibrahim HA. Determinants of High Breastfeeding Self-Efficacy among Nursing Mothers in Najran, Saudi Arabia. Nutrients. 2023; 15(8):1919. https://doi.org/10.3390/nu15081919.

Reviewer 2 Report

Comments and Suggestions for Authors

The manuscript presents a well-conducted study with clear methodology and comprehensive results. The introduction sets the stage effectively, and the discussion provides meaningful interpretation. With minor improvements in the clarity of objectives, transitions, and more detailed explanations in certain sections, the manuscript would be further strengthened. See specific comments below

1.      Introduction: The manuscript begins with a clear introduction, emphasizing the importance of breastfeeding (BF) as recognized by the World Health Organization introduction smoothly transitions into the context of breastfeeding practices in Spain, supported by data from the 2017 National Health Survey. The introduction could benefit from a more detailed explanation of the specific objectives and hypotheses of the study. This would provide a clearer roadmap for the reader.

2.      Method: The sample selection process, specifically the method of intentional sampling, could be elaborated to provide more context on why these particular PHCs were chosen. Moreover, a brief explanation of the rationale behind using the IIFAS and its relevance to the study objectives would enhance the reader's understanding.

3.      Results: Some results sections could benefit from additional narrative explanations to provide more context and interpretation of the data presented in the tables. Highlighting key findings more explicitly in the text would help emphasize the most important results for the reader.

4.      Some results sections could benefit from additional narrative explanations to provide more context and interpretation of the data presented in the tables.

5.      Highlighting key findings more explicitly in the text would help emphasize the most important results for the reader. However, it could benefit from more specific recommendations for future research, particularly regarding prospective study designs to corroborate the findings. A more detailed exploration of the potential mechanisms underlying the observed associations would enhance the depth of the discussion.

Author Response

REVIEWER 2.

The manuscript presents a well-conducted study with clear methodology and comprehensive results.The introduction sets the stage effectively, and the discussion provides meaningful interpretation. With minor improvements in the clarity of objectives, transitions, and more detailed explanations in certain sections, the manuscript would be further strengthened.

Dear reviewer,

Thank you very much for your kind words. This encourages us to continue working in this direction.

See specific comments below.

1. Introduction: The manuscript begins with a clear introduction, emphasizing the importance of breastfeeding (BF) as recognized by the World Health Organization introduction smoothly transitions into the context of breastfeeding practices in Spain, supported by data from the 2017 National Health Survey. The introduction could benefit from a more detailed explanation of the specific objectives and hypotheses of the study. This would provide a clearer roadmap for the reader.

We consider it very important that you have conveyed these comments to us, as it allows us to understand that our theoretical framework has been constructed correctly.

Regarding your request, the specific objectives of the study and the operational hypothesis are detailed in greater depth. In the new version of the manuscript, it appears as follows: “Based on the aforementioned points, it could be expected that a BFHI hospital would have a high proportion of mothers who exclusively breastfeed during hospitalization and continue breastfeeding after discharge. If this is not the case, it is important to understand the factors that may influence these decisions. Our objectives were to determine the percentage of mothers who practiced exclusive BF during hospitalization and which factors related to mothers could influence the degree of exclusive breastfeeding during hospitalization, as well as to assess breastfeeding mothers' attitudes towards breastfeeding.”

2. Method: The sample selection process, specifically the method of intentional sampling, could be elaborated to provide more context on why these particular PHCs were chosen.

Thank you very much. The sample size was calculated a priori before initiating the study. For sample size calculation, considering that there were 1948 births in our healthcare area during the year 2023, and assuming the percentage of exclusive breastfeeding at discharge was estimated (based on previous literature) to be 87.5% [1], and using calculation based on a binomial distribution, a sample size of at least 62 women was required to estimate a proportion of 87.5%, with an absolute error of 10%, and for a confidence level of 95%. In our study, we recruited a total of 64 women.

Once we set the sample size a priori, we proceeded to select the primary healthcare centers (PHCs) that would participate. Starting from a list of all primary healthcare centers in the healthcare area (74 in total), we randomly selected 5 primary healthcare centers to recruit participants. The five selected centers mentioned in the manuscript were Concepción Arenal PHC and Vite PHC (urban), and Boqueixón PHC, O Pino PHC, and Touro PHC (rural).

After selecting the participating healthcare centers, women attending a routine check-up at the center and meeting the selection criteria were invited to participate in the study. All 64 women invited agreed to participate; there were no refusals. Recruitment was stopped upon reaching 64 participants, as we had reached the target sample size.

This information on sample size is included in the materials and methods section in the new version of the text. In the revised manuscript, it is presented as follows: “The sample size was calculated prior to starting the study. Based on the number of births in our healthcare area in 2023, the percentage of exclusive breastfeeding at discharge was estimated at 87.5% [25]. Using a binomial distribution, a sample of at least 62 women was needed, with a 10% absolute error and a 95% confidence level. Once the sample size was determined, primary healthcare centers (PHCs) were randomly selected. From a list of 74 PHCs, 5 were randomly chosen: 2 urban (Concepción Arenal PHC and Vite PHC) and 3 rural (Boqueixón PHC, O Pino PHC, and Touro PHC). Women meeting the selection criteria were invited to participate during routine check-ups through purposive sampling.”

[1] (ref. 25 in manuscript) García García N, Fernández Gutiérrez P. Conocimientos y actitudes de las madres ante la lactancia materna en un hospital IHAN. Metas Enferm 2018; 21(1):50-8. doi:10.35667/MetasEnf.2019.21.1003081174

Moreover, a brief explanation of the rationale behind using the IIFAS and its relevance to the study objectives would enhance the reader's understanding.

Thank you for your input. The IISFAS-s questionnaire was used because it is a valid and reliable tool for measuring maternal attitudes towards breastfeeding. This allows us to estimate, based on the scores of this questionnaire, the effect that a positive attitude towards breastfeeding has on maintaining exclusive breastfeeding during hospitalization. In our sample, we did not observe any women categorized as having a positive attitude towards breastfeeding. 21 (32.8%) women exhibited a neutral attitude, while 43 (67.2%) showed a positive attitude towards breastfeeding.

This information is included in the new version of the manuscript. It now appears in the methodology section as follows: "“IIFAS-s questionnaire was used because it is a valid and reliable tool for measuring mothers' attitudes towards breastfeeding. This allows us to estimate, based on the scores of this questionnaire, the effect that a positive attitude towards breastfeeding has on maintaining exclusive breastfeeding during hospitalization.”

The following sentence is included in the results section: “In our sample, we did not observe any women who fell into the negative attitude towards breastfeeding category. Twenty-one (32.8%) women exhibited a neutral attitude, and forty-three (67.2%) showed a positive attitude towards breastfeeding.”

3. Results: Some results sections could benefit from additional narrative explanations to provide more context and interpretation of the data presented in the tables. Highlighting key findings more explicitly in the text would help emphasize the most important results for the reader.

Thank you very much for your contribution. Several points of the analysis have been explained. You can observe those changes in the results section. We hope that interpretation is now easier.

4. However, it could benefit from more specific recommendations for future research, particularly regarding prospective study designs to corroborate the findings. A more detailed exploration of the potential mechanisms underlying the observed associations would enhance the depth of the discussion.

Thank you very much for your recommendation. Following your guidance, we have expanded and delved deeper into the discussion of the results. New text has been incorporated into the new version of the manuscript. Now, in the discussion section, the following is included in the second paragraph:: “Despite giving birth in a BFHI hospital, 35.9% of the women did not receive assistance or information about breastfeeding from the healthcare professionals who cared for them during their stay. Among those who did receive assistance, a high percentage considered it could be improved. A similar situation occurred at the primary healthcare level. This could be explained by the need to improve training activities for healthcare professionals.”

In the third paragraph, the following is incorporated: “Two areas have been identified where specific interventions could be focused on the healthcare staff who care for women during their hospital stay. By concentrating training interventions on promoting skin-to-skin contact within the first 30 minutes, as well as equipping hospitals with more humanized and open neonatal units that encourage activities like the kangaroo mother care technique and skin-to-skin contact, the likelihood that mothers will choose exclusive breastfeeding during their stay would increase. Additionally, the duration of breastfeeding is likely to extend once they are discharged from the hospital. There is evidence that addressing the two identified areas would achieve these goals [42,43].”

In the fourth paragraph: “In this study, we did not observe any women who fell into the negative attitude towards breastfeeding category and 67.2% showed a positive attitude towards breastfeeding. This result is much higher than those found in previous studies [46,47]. It should be taken into account when interpreting these results that the women who participated did so after their hospital stay, while they were being followed up by pediatric nurses at their PHCs. This may be influenced by the fact that the women received advice at that time from their pediatric nurse and breastfeeding support groups.”

A new paragraph is added among the study's strengths, arising from the changes introduced due to the reviewers' recommendations: “Another strength of this study is that it allows us to identify areas for improvement or gaps to target and focus interventions on, both for mothers and healthcare professionals, as well as the healthcare system. This is consistent with similar studies [48].”

Additionally, in accordance with the reviewers' suggestions, the conclusion section is modified: “In our study, exclusive breastfeeding during hospitalization is associated with having previous children and prior breastfeeding experience. It also appears to be linked to the likelihood of the newborn not being admitted to neonatal units. This association is influenced by the mother's age, breastfeeding information provided during hospitalization, and skin-to-skin contact within the first 30 minutes. Exclusive breastfeeding during hospitalization could be improved by increasing healthcare staff training and encouraging their involvement in achieving goals according to BFHI criteria. Future research with prospective designs is needed to measure the effect of multifaceted interventions focused on our findings, in order to estimate the association with exclusive breastfeeding during hospitalization and its duration after discharge.”

Due to all the changes prompted by the reviewers' suggestions, new bibliographic references had to be included. It are:

42. Song JT, Kinshella MW, Kawaza K, Goldfarb DM. Neonatal Intensive Care Unit Interventions to Improve Breastfeeding Rates at Discharge Among Preterm and Low Birth Weight Infants: A Systematic Review and Meta-Analysis. Breastfeed Med. 2023;18(2):97-106. doi:10.1089/bfm.2022.0151.

43. Renfrew MJ, Dyson L, McCormick F, Misso K, Stenhouse E, King SE, Williams AF. Breastfeeding promotion for infants in neonatal units: a systematic review. Child Care Health Dev. 2010;36(2):165-78. doi:10.1111/j.1365-2214.2009.01018.x.

46. Alkhaldi SM, Al-Kuran O, AlAdwan MM, Dabbah TA, Dalky HF, Badran E. Determinants of breastfeeding attitudes of mothers in Jordan: A cross-sectional study. PLoS One. 2023 May 5;18(5):e0285436. doi:10.1371/journal.pone.0285436.

47. Han FL, Ho YJ, McGrath JM. The influence of breastfeeding attitudes on breastfeeding behavior of postpartum women and their spouses. Heliyon. 2023;9(3):e13987. doi:10.1016/j.heliyon.2023.e13987.

48. Al-Thubaity DD, Alshahrani MA, Elgzar WT, Ibrahim HA. Determinants of High Breastfeeding Self-Efficacy among Nursing Mothers in Najran, Saudi Arabia. Nutrients. 2023; 15(8):1919. https://doi.org/10.3390/nu15081919.

Round 2

Reviewer 1 Report

Comments and Suggestions for Authors

Dear authors, thank you for your response to the review of the manuscript and for putting a lot of work into preparing a revised version.

In the revised version of the manuscript, the authors have incorporated all my comments.